# Well-Defined Nanostructures by Block Copolymers and Mass Transport Applications in Energy Conversion

**DOI:** 10.3390/polym14214568

**Published:** 2022-10-28

**Authors:** Shuhui Ma, Yushuang Hou, Jinlin Hao, Cuncai Lin, Jiawei Zhao, Xin Sui

**Affiliations:** College of Materials Science and Engineering, Qingdao University, Qingdao 266071, China

**Keywords:** block copolymer, self-assembly, nanochannels, membranes, energy conversion

## Abstract

With the speedy progress in the research of nanomaterials, self-assembly technology has captured the high-profile interest of researchers because of its simplicity and ease of spontaneous formation of a stable ordered aggregation system. The self-assembly of block copolymers can be precisely regulated at the nanoscale to overcome the physical limits of conventional processing techniques. This bottom-up assembly strategy is simple, easy to control, and associated with high density and high order, which is of great significance for mass transportation through membrane materials. In this review, to investigate the regulation of block copolymer self-assembly structures, we systematically explored the factors that affect the self-assembly nanostructure. After discussing the formation of nanostructures of diverse block copolymers, this review highlights block copolymer-based mass transport membranes, which play the role of “energy enhancers” in concentration cells, fuel cells, and rechargeable batteries. We firmly believe that the introduction of block copolymers can facilitate the novel energy conversion to an entirely new plateau, and the research can inform a new generation of block copolymers for more promotion and improvement in new energy applications.

## 1. Introduction

Two or more thermodynamically incompatible polymer blocks with diverse physical and chemical properties are covalently bonded together to form so-called block copolymers (BCPs) [1]. In the 1950s, the discovery of the surfactant Pluronic (PEO-b-PPO-b-PEO) [2,3,4] first attracted the attention of scientists. The mass production of BCP received a boost in 1956 due to the invention of living anionic polymerization [5]. Subsequently, other controlling polymerization techniques can also be used to synthesize BCP, such as living cationic polymerization [6], atom transfer radical polymerization (ATRP) [7,8,9,10,11], and reversible addition-fragmentation chain transfer polymerization (RAFT) [12,13]. Edwards’ self-consistent field theory (SCFT) [14] provided a premium theoretical tool for posterior simulations to explore the phase behavior of BCP. The foundations laid by these pioneers [15,16] have greatly facilitated the advancement of materials science, technology, and theory [17]. The chemical properties of various BCP blocks tend to vary with amenability to addition. To exploit the controllable properties of BCP, scientists have combined theoretical derivation with experimental practice to modify the structure of BCP for desired applications. This requires a thorough understanding of the factors that govern the nanostructure of BCP to regulate it according to the actual needs.

The BCP system can stand out and become the focus of research due to its incomparable self-assembly ability [18], and BCP can undergo accurate microphase separation in the range of 10–100 nm, which expands a new method to prepare nanodevices. The self-assembly of BCP forms rich nanostructures due to the difference in structure, block ratio, and sequences [19]. Surface structure and nanostructure are the keys to improving the properties in applications such as electronic lithography [20], organic photovoltaics [21], nanofiltration [22], and ultrafiltration [23], where electron transport or surface features dictate the performance of the devices. BCP has the potential to boost the properties of these nanodevices [24], which can result in their flourishing applications in different areas due to unique nanoscale customizable template nanostructures [25]. This review intends to highlight the effective exploitation and performance enhancement of BCP for novel energy sources instead of repeating the aforementioned applications. Our main focus will be on the construction and application of BCP-based nanostructured materials in mass transport for energy conversion. The massive exploitation and usage of nonrenewable resources, such as oil and coal, has become a burden on the environment [26]. In recent years, global warming has been responsible for frequent natural disasters [27,28], which have resulted in an irresistible surge to limit the traditional energy methods that incur huge carbon emissions and foster innovative and efficient clean energy sources. Concentration cells, fuel cells, and rechargeable batteries that can convert chemical energy into electricity without carbon emissions are new energy sources with the potential to replace traditional energy sources in the foreseeable future [29]. Using membranes for mass transportation is a key part to obtain high power efficiency from the cells of these devices related to new energy conversion processes [30], which makes the development of a high-performance battery diaphragm very important.

Because the so-called structure determines the properties, we devote a fairly lengthy section to the nanostructure of BCP before presenting their properties relevant for various practical applications. First, we briefly analyze the relevance of three vital parameters in the microphase separation of BCP. Second, various factors affect the self-assembly structure of BCP, such as the chemical structure and external conditions during the assembly process. We elaborate on the effect of various factors on the regulation of self-assembled structures, such as the chemical structure, solvent types, polymer solution concentration, nonsolvent, external field conditions, and additive aspects. Well-defined nanostructures with high order and high density can be obtained by this bottom-up self-assembly method of BCP, which can be used as nanochannel templates for mass transport. Finally, as the highlight of this paper, we investigate the application of BCP-based membranes in energy conversion, such as concentration cells, fuel cells, and rechargeable batteries. Finally, we conclude with futuristic perspectives. We hope that this review can provide a theoretical basis for how to regulate the nanostructure of polymers and offer ideas to further promote the application of block copolymers in new energy fields.

## 2. Microphase Separation of Block Copolymer

Spontaneous phase separation occurs in BCP due to repulsion between chemically and thermodynamically incompatible blocks. At the macroscopic scale, the phase separation between blocks cannot occur due to the presence of covalent bonds between the blocks of BCP [31]. Instead, in the macromolecular-length scale, phase separation can only occur at the microscopic scale, i.e., microphase separation [32]. Due to the microphase separation, there is structural arrangement of BCP at the macroscopic scale, which is known as the self-assembly phenomenon [33] and responsible for the generation of new boundaries between blocks and can result in various assembly structures.

Three significant parameters are responsible for the microphase separation between the blocks: (1) the relative volume fraction *f* of each block, which is used to characterize the microscopic composition of BCP; (2) the Flory–Huggins parameter *χ*, which is used to characterize the interaction between the blocks [34,35] and is inversely proportional to the temperature change; (3) the total degree of polymerization *N* of the polymer. Among them, *χN*, which represents the phase separation strength, plays a decisive function in the separation state of BCP [36]. Increasing the temperature or decreasing *χN* gradually decreased the incompatibility between the blocks. Molecular-level mixing changes polymeric molecular chain from the stretched chain state with obvious interfaces and ordered arrangement to the disordered Gaussian chain state, as shown in Figure 1a, which results in an order-to-disorder transition (ODT) [37] of the polymer. The critical *χN* value for the occurrence of ODT is approximately 10.5. Electron microscopy can be used to clearly observe the ODT transition process of some BCPs with lower molecular weights, as shown in Figure 1b, which shows TEM images of poly(styrene)-b-poly(imide) (PS-b-PI) before and after undergoing the ODT process.

Due to the introduction of new theories such as the thermal up-and-down effect and mean-field theory, the improved self-consistent field theory (SCFT) can more accurately calculate the BCP phase behavior by scientists [38]. The theoretical phase diagram of the AB diblock copolymer [39] based on self-consistent mean-field (SCMF) theory showing the changes in phase behavior with increasing or decreasing values of *f_A_* and *χN* in a concrete manner is shown in Figure 1c (left). However, the experimental phase diagram deviates from the theoretical phase diagram because there are various practical and objective factors. Figure 1c (right) shows the experimental phase diagram of the PI-b-PS copolymer obtained by Bates et al. Both experimental and theoretical phase diagrams of PI-b-PS are qualitatively consistent with the variation occurring on the quantitative scale. First, the critical *χN* value for ODT is approximately 20, with the loss of symmetry for the phase diagram at approximately *f_A_* = 1/2. Instead of the CPS phase, a mesostable porous lamellar PL phase appears between C and L regions (calculations confirmed the mesostability of this phase later on). The difference in the PI and PS blocks from the theoretically assumed AB blocks is the main reason behind these deviations. The molecular morphology, properties, and interactions between PI and PS are responsible for the difference in experimental and theoretical values. In the experimental phase diagram, the ODT transitions can directly proceed in both the disordered D region and ordered regions that only occur at the critical point in the theoretical phase diagram, and this phenomenon can be mainly attributed to the effect of the thermal up-and-down phenomenon. With an increasing number of blocks, the phase behavior of block copolymers becomes more complicated [40,41,42].

**Figure 1 polymers-14-04568-f001:**
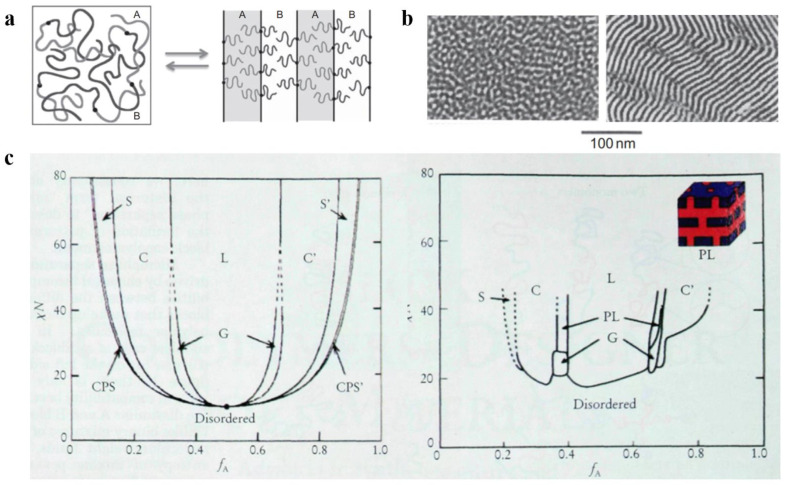
(**a**) Molecular chain distribution of AB-type diblock copolymer before and after the ODT [37]. (**b**) TEM images of PS-b-PI before and after the ODT [37]. (**c**) Theoretical phase diagrams predicted by the self-consistent field theory of the linear AB diblock copolymer and the actual phase diagram (spherical (S, S’), cylindrical (C, C’), bicontinuous gyroidal (G), and lamellar (L)) obtained from the PS-b-PI [39].

## 3. Regulation of the Self-Assembled Structure of the Block Copolymer

### 3.1. Influence of the Molecular Structure on the Self-Assembled Structure

The structural morphology of BCP films is our main interest. The self-assembly of BCP film is affected by many factors [43,44,45] which can be divided into molecular structure factors and assembly condition factors. The most fundamental strategy to regulate the nanoscale morphology of BCP is to adjust its molecular structure, since the BCP structure determines the self-assembly properties. Above the ODT transition curve, the thermodynamically stable nanostructures of BCP mainly consist of spherical S, cylindrical C, bicontinuous gyroidal G, and lamellar L geometries. As shown in Figure 2a, a change in volume fraction f of the AB block changes the morphology of the AB diblock copolymer [46]. Depending on whether f_A_ < 0.5 or f_B_ < 0.5, block A is dispersed within the continuous phase composed of block B or vice versa, and with increasing f_A_, the transformation of S→C→G→L occurs, while the same S’→C’→G’→L transition occurs with increasing f_B_. The size of each block in each form is determined by the molecular weight and compatibility of the blocks. With increasing molecular weight of the polymer, the blocks become segregated with increasing distance between the domains, and these factors make it challenging to undertake studies of high molecular weight BCPs [47].

The structure of BCP is also characterized by an essential parameter known as polydispersity index (PDI) [48] of molecular weight, which is calculated as PDI = M_n_/M_w_ [49]. The BCPs obtained from the polymerization of reactive anions are monodisperse [50]. However, in recent years, with the emergence of novel methods for synthesizing BCPs, most of the obtained BCPs have been polydisperse [51]. It is necessary to study the effect of broadening the molecular-weight distribution on the phase behavior of BCPs. With the concept of maintaining thermodynamic equilibrium to the maximum possible intent, Ruzette et al. [52] investigated the self-assembly behavior of triblock copolymer ABA with varying molecular-weight distributions, with narrower dispersion of block B (poly (butyl acrylate) (PBA)) and wider dispersion of block A (poly (methyl methacrylate) (PMMA)). The resulting transparent BCP membranes are homogeneous. The TEM characterization reveals the possibility of forming classical S, C, G, and L structures. However, a highly curved interface between the blocks and the PMMA block was formed mainly due to the shift of the interface to release the stretching energy of the molecular chains.

Lynd et al. [53] investigated the effect of molecular-weight distribution on the occurrence of ODT for two different BCPs prepared of polydispersed poly (ethylene-acrylamide)-b-poly (DL-propyleneglycol) (PL) and polystyrene-b-polyisoprene (SI). Ultimately, during the occurrence of ODT, increasing polydispersity in blocks with fewer (*f* < 0.5) or more (*f* > 0.5) components decrease or increase in the critical *χN* value, respectively. Widin et al. [54] reported the structural effects of a low-dispersion block polystyrene (PS) and a high-dispersion block polybutylene (PB) on the triblock copolymer SBS. As shown in Figure 2b (left), the molecular chain of highly dispersive block PB is stabilized by the low dispersive PS block in the middle position. Finally, similar experimental results to those of Ruzette were obtained, where the phase separation interface was bent toward the polydisperse segment because the polydisperse block has a smaller filling volume than the monodisperse block of the same molecular weight (the phase interface bending process is shown in Figure 2b (right)). The polydisperse B-block also reduces the critical *χN* value for ODT and increases the position of the lamellar phase window. The investigations show that by playing with the polydispersity of a particular block in BCP, the microphase separation state of BCP and its thermodynamic stability can be regulated.

**Figure 2 polymers-14-04568-f002:**
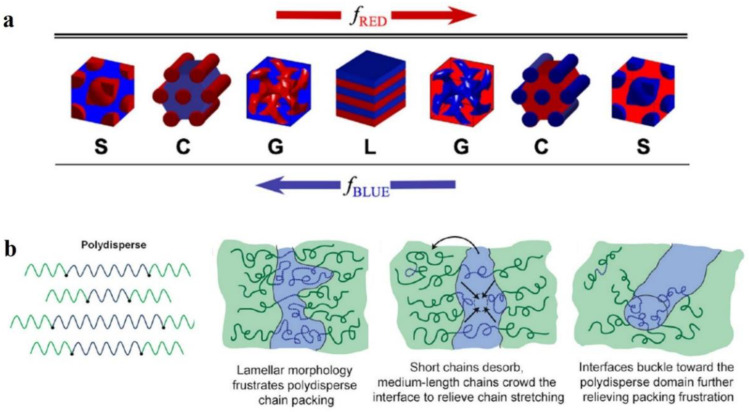
(**a**) Four equilibrium-ordered state transitions in diblock copolymers determined by relative volume fraction *f*. Spherical (S), cylindrical (C), bicontinuous gyroidal (G), and lamellar (L) [46]. (**b**) Schematic diagram of polydispersed B-blocks in ABA-type triblock copolymers leading to distortion of block microdomain interfaces [54].

### 3.2. Influence of Assembly Conditions on the Self-Assembled Structure

Normally, in addition to the interactions between polymer molecules, external film formation conditions can also affect the nanostructure. Hence, the type of solvent, polymer solution concentration, nonsolvent, additives, and external field conditions during the film formation process can affect the structure of BCP.

#### 3.2.1. Solvents to Prepare the Membranes and Self-Assembly

The selectivity of solvents toward different blocks significantly affect the self-assembled structures. Hanley et al. [55] dissolved PS-b-PI in bis(2-ethylhexyl) phthalate (DOP), which is a nonselective solvent for PS and PI, di-n-butyl phthalate (DBP), and diethyl phthalate (DEP) solvents, which are sequentially more selective for the PS block, and tetradecane (C14), which is a selective solvent for the PI blocks, to observe their phase behavior by small-angle X-ray scattering (SAXS) (Figure 3a). The PS-b-PI membrane exhibited the transformation of G→hexagonal filled C→D, which, when dissolved in different solvents, exhibited different nanostructures. After dissolution in DBP, the phase transition process of PS-b-PI was L→G’→C’→S’. The phase behavior in DEP, which is more selective for PS, like DBP, was similar to that of DBP but with higher *T_ODT_*. In contrast, in C14, for the PI selective solvent, the G’→C’→S’ ordered phases were the only occurring transitions because enhancement of segregation between microdomains by the selective solvents improve the stability of the ordered phase, which results in the appearance of more ordered phase behavior. The formation and closure of pores in BCP nanostructures can also be regulated by selective solvents [56]. The spontaneously formed poly(styrene)-b-poly (2-vinyl pyridine) (PS-b-P2VP) membrane is a dense, nonporous film. However, when dissolved in P2VP selective solvent (e.g., ethanol), the P2VP blocks dissolve to form a membrane with round or elliptical pores surrounding the core of the PS blocks. With increasing time or temperature, these pores grow into a columnar network and finally form three-dimensional interconnected pores. This pore formation process is often reversible, and immersion of the porous membrane in a PS block selective solvent (e.g., cyclohexane) closes the pores to restore the dense, nonporous membrane.

In addition, dissolution of the self-assembled structure of BCPs in mixed solvents should be explored. Yi et al. [57] investigated the nanostructure of poly(styrene)-b-poly (4-vinyl pyridine) (PS-b-P4VP) by adding it to N, N-dimethylformamide, DMF, (selective for P4VP block, and slow volatilization), tetrahydrofuran, THF, (selective for PS, faster volatilization), and their mixtures. In pure DMF solvent, a nanoscale spherical structure was obtained for PS-b-P4VP, a dense and smooth surface was obtained in pure THF, while the mixed solvent yielded a rod-like aggregated structure where the diameter of the nanospheres was similar to that observed in DMF. When polymers are dissolved in a mixture of solvents with different solubilities and volatilities, the preferential evaporation of one solvent leads to a concentration gradient within the BCP. This concentration gradient leads to phase separation of the BCP, the so-called evaporation-induced phase separation (EIPS) [58] of the solvent.

In addition, the concentration of the polymer solution is an important factor that affects the nanostructure of BCP. Jiang et al. [59] was the first to simulate the membrane formation of a double hydrophobic poly(styrene)-b-poly (methyl methacrylate) (PS-b-PMMA). The effect of the polymer concentration on the morphology of polymer molecules was also investigated in the polymer concentration range of 10–55%. Figure 3b clearly shows the difference in morphologies corresponding to different polymer concentrations. For low polymer concentrations, microphase separation just begins to occur, and the formation of a continuous phase in the system under this condition is not possible because there is not a sufficient BCP concentration, which forms a spherical structure at lower concentrations. With increasing polymer concentration, a structure can be observed because a continuous phase forms by BCP. With a further increase in concentration, porous membranes with elliptical pores start to form. When the concentration increases to approximately 30%, the elliptical pores change to circular pores. From there on, a further increase in polymer concentration only decreases the pore size within the porous membrane and membrane wall thickening. When the polymer concentration reaches approximately 45%, the polymer shows an irregular sponge-like, cross-linked structure, and finally, a continued increase in polymer concentration causes the pores to plug and results in a continuous structure of the polymer. This experiment indicates that, within a certain concentration range, the polymer concentration can be adjusted to achieve different polymer morphologies and obtain porous membranes with ideal pore size and stability.

Formation of the BCP membrane by selective solvents requires three steps: solvent uptake, swelling, and solvent evaporation drying. The effects of the first two steps on the BCP have been described in the previous section, while the evaporation rate of the solvent has been found to affect the structural orientation of BCP. It has been proven that fast solvent evaporation leads to a vertical orientation of the internal structure [60], while slower solvent evaporation rates cause parallel structural orientation or structures with mixed orientation [61]. The evaporation rate determines the propagation of phase separation from the membrane surface to the interior. Similar conclusions were obtained by Phillip et al. [62] when comparing the morphology of poly(styrene)-b-poly (D, L-lactide) (PS-b-PLA) under rapid solvent evaporation and slow evaporation (Figure 3c). Because ultrafast solvent evaporation makes steep concentration gradients rapidly propagate the vertical direction of the membrane surface, the cylindrical structure orients along the surface’s normal direction.

**Figure 3 polymers-14-04568-f003:**
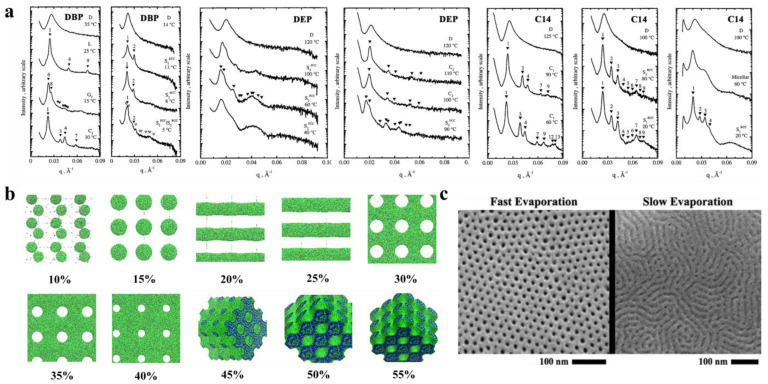
(**a**) Representative SAXS profiles as a function of temperature for DBP, DEP, and C14 solutions [55]. (**b**) Nanoscale morphology diagram of PS-b-PMMA porous membrane simulated by DPD when dissolved in tetrahydrofuran solvent at different polymer concentrations. PMMA is shown in green, and PS is shown in blue [59]. (**c**) SEM images of different morphologies formed by PS-b-PLA at varied solvent evaporation rates [62].

During the self-assembly process via solvent vapor annealing (SVA), the solvents are found to have a significant influence. For a given polymer system, the choice of different solvents can result in different nanostructures [63,64,65]. Peng et al. [66] investigated the effect of SVA on the morphology of poly(styrene)-b- poly (ethylene oxide) (PS-b-PEO). As shown in Figure 4a, the nanostructure of the PS-b-PEO membrane obtained by conventional casting at room temperature was a disordered and irregular worm-like morphology, while the PS-b-PEO membrane resulting from 5 min of steam annealing treatment with toluene (a selective solvent for PS) was transformed into a highly ordered arrangement of hexagonally filled perpendicular cylindrical microdomains.

However, due to preferential wetting of the surface, a thermodynamically preferred morphology should be formed with a cylindrical orientation parallel to the surface instead of perpendicular to the surface, as experimentally obtained in [67]. The formation mechanism of this morphology has not been understood, and various researchers have attempted to use computer simulations or experimental investigations to explain this phenomenon [68,69,70]. Phillip et al. [71] analyzed the evolution of the internal morphology of PS-b-PLA during SVA. Theoretical analysis and experimental verification found that the generation of such cylinders evolved from spherical intermediates nucleated at the vapor/polymer membrane interface due to the rapid evaporation of solvent, which created a concentration gradient in the BCP, and the perpendicular orientation of the intermediates with respect to the surface and epitaxial growth into cylinders. However, too-fast evaporation of the solvent does not allow sufficient time for nucleation. Lin et al. [72] used TEM analysis to demonstrate that the length of this ordered arrangement could reach the level of membrane thickness. In other words, SVA can achieve a directional ordered arrangement perpendicular to the membrane surface throughout the membrane, which is of great significance for the development of certain ion- or proton-conducting membranes or water-permeation membranes with controlled pore sizes. Kim et al. [73] demonstrated that SVA yielded a more ordered, structurally stable membrane than those obtained by conventional methods by imparting the molecular chain mobility, shielding the surface energy of the two blocks at the vapor/BCP interface and the interfacial energy between polymer and substrate, and rapidly eliminating defects in the BCP structure (Figure 4b).

When the blocks in BCP strongly interact, more variables in the SVA process will affect the structure [65,74,75]. For instance, Park et al. [76] proposed structural changes in a PS-b-PMMA membrane by choosing a selective or neutral substrate (different substrate treatments with the generated SEM images of varying morphologies are shown in Figure 4c). The membranes on neutral substrates exhibit a cylindrical structure perpendicular to the surface at the top (near the air) and bottom (near the substrate). For the SVA process on selective substrates, the top structure of the BCP membrane was unaffected, exhibiting a cylindrical structure perpendicular to the surface, but the bottom structure changed to a cylindrical structure parallel to the substrate. This result demonstrates the possibility of tailoring the pattern and structure of the BCP membrane by adjusting the membrane–substrate interaction. Cheng et al. [77] investigated various process parameters during the SVA process and ultimately demonstrated that the temperature of the substrate and vapor were also key factors in controlling the structural morphology of BCP membranes.

However, the ordered BCP membrane obtained by the selective solvent SVA method is only short-ranged. It has been shown that long-range ordering of BCP membrane in the lateral direction can be obtained via annealing with nonselective solvent (solvent with the same selectivity for each block and no specific interactions) vapors during SVA [78]. Bosworth et al. [79] obtained a long-range ordered BCP membrane by annealing poly(α-methylstyrene)-block-poly(4-hydroxystyrene) in acetone (selective solvent) vapor, which resulted in a spherical structure and tetrahydrofuran (nonselective solvent) vapor, forming a columnar phase parallel to the surface. This is due to the fact that the nonselective solvent is only uniformly distributed between the molecular chains and same selectivity for each block. It increases only their mobility without any other interactions, resulting in spontaneous formation of a cylindrical orientation parallel to the membrane plane with the lowest surface energy. SVA overcomes the shortcomings of conventional thermal annealing and enables highly ordered rearrangement of BCP in a relatively short time. By reducing the effective *T_g_*, it weakens the interactions between chains and between chains and substrates to increase the mobility of polymer chain [80].

**Figure 4 polymers-14-04568-f004:**
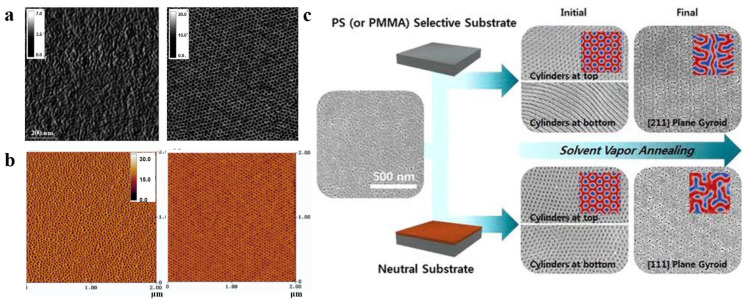
(**a**) AFM phase images (1 × 1 μm^2^) of as-cast conventional PS-b-PEO film and PS-b-PEO film annealed in PS-selective toluene vapor for 5 min [66]. (**b**) AFM phase diagrams of PS-b-PEO films obtained by spin-coating method and annealed in neutral benzene vapor for 48 h [73]. (**c**) SEM images of the (111) and (211) planes produced by solvent vapor annealing of PS-b-PMMA films in neutral substrate and selective substrate [76].

#### 3.2.2. Nonsolvent Approaches for Constructing Nanostructure

Researchers have also investigated the effect of nonsolvent-induced phase separation (NIPS) [81] of BCPs intensively [82,83,84]. The polymer is dissolved in solvent to form a homogeneous solution, followed by the slow addition of reagents which are more soluble in the solvent compared to the polymer (called extractants) to extract the solvent. The resulting two-phase structure with the polymer as continuous phase and solvent as the dispersed phase can lead to a polymer with a certain pore structure after the removal of solvent. Plisko et al. [85] modified poly (ethylene glycol)-b-poly (propylene glycol)-b-poly (ethylene glycol) (Pluronic F127) by the NIPS method employing polysulfone (PSF), and the polymer membrane obtained by the NIPS method was found to be porous in comparison to the dense polymer membrane prepared by the EIPS method (the surface structure of the porous film is shown in Figure 5a). Similarly, Gu et al. [86] fabricated porous PI-b-PS-b-P4VP membrane with sponge-like structure using the NIPS method, and Figure 5b shows the cross-sectional structure of the sponge-like porous film. NIPS is the most commonly used method for the preparation of membranes for separation [87,88], but the pore size and distribution of pores on the surface of NIPS-formed membranes are irregular and not well-controlled [89,90], and presence of some large pores can lead to material defects resulting in reduced mechanical strength of the membrane. This feature is responsible for limiting the application of BCP membranes made by the NIPS method in a great way.

In order to overcome the shortcoming mentioned above, a new NIPS strategy has emerged in the recent years: a short self-assembly is formed via short solvent evaporation after pouring the BCP into a solvent or mixed solvent, and then it was immersed in a nonsolvent for NIPS [91,92]. By this method, a monolithic, asymmetrically structured membrane, as shown in Figure 5c,d, can be formed [93]. Self-assembly results in the formation of highly ordered, dense pores in the upper thin selective layer which are uniformly perpendicular to the membrane surface [94]; pores of the lower layer which exhibit a sparse, disordered sponge-like interconnected pore channel are formed by NIPS [95]. This method of combining self-assembly (S) of BCP with nonsolvent-induced phase separation (NIPS) is named SNIPS [96,97]. As the first step of membrane formation via SNIPS is dissolution of BCP for self-assembly, different solvent parameters that affect the nanostructure (different selective solvents, mixed solvents, and solvent volatilization rate) discussed in the previous section also affect the membrane structure formed by the SNIPS method [98,99,100].

**Figure 5 polymers-14-04568-f005:**
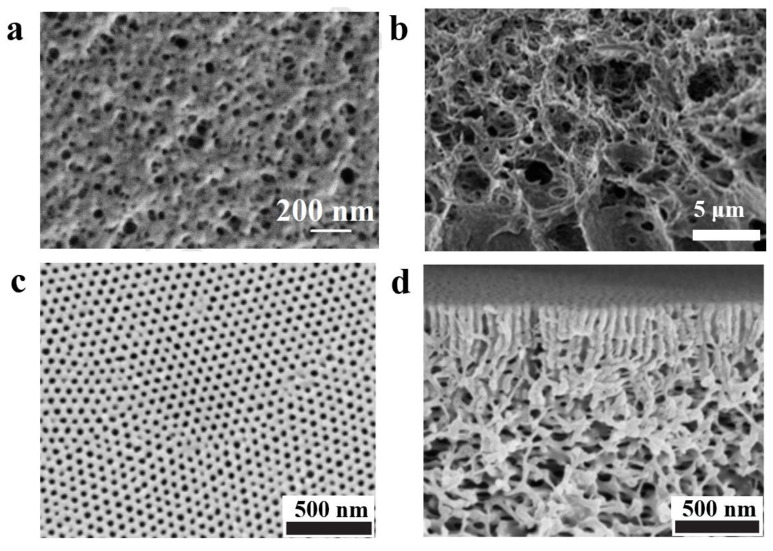
(**a**) SEM image of the surface of porous ultrafiltration membrane formed by the NIPS method for Pluronic F127 [85]. (**b**) SEM image of cross-section of multiporous sponge-like structure PI-b-PS-b-P4VP formed by NIPS method [86]. SEM images of the surface (**c**) and cross section (**d**) of PS-b-P4VP membrane obtained by SNIPS [93].

#### 3.2.3. The Influence of External Field on Self-Assembled Structure of BCP

Thermal annealing has proved to be one of the efficient methods for preparing long-range ordered BCP membranes [101,102,103,104], and temperature is an effective and adjustable experimental condition during the process of self-assembly. Thermal annealing refers to the heating of BCP above its glass transition temperature T_g_ [105] for allowing the molecular chains within the polymer to self-assemble to the state closest to thermodynamic equilibrium [106,107,108]. Stehlin et al. [109] used AFM to observe the morphological variation of PS-b-PMMA at 180 °C and 200 °C (well above its T_g_). Their resultant AFM images are shown in Figure 6a. It illustrates that thermal annealing results in ordered arrangement of the molecular chains, and the orderliness of PS-b-PMMA structure at 200 °C is higher than that of 180 °C for the same time, which indicates that the higher the annealing temperature, the faster the order of the arrangement. However, the BCP membrane produced by conventional thermal annealing techniques often shows structural defects, and the range of order regulation is limited, and for a thick membrane, ordering of the molecular chains over a large area cannot be obtained [110]. Moreover, for some polymers, the spacing between T_g_ and the viscous flow temperature T_f_ is small, which may not be sufficient for the rearrangement of molecular chains to take place within a short time.

In some systems, a combined method of the thermal annealing and solvent annealing methods can be used [111]. Kim et al. [112] experimentally investigated PS-b-P4VP membrane in chloroform vapor (the schematic diagram is shown in Figure 6b). Since chloroform is selective for the PS block, the PS block is pulled toward the outside of the membrane, and the P4VP block forms multilayer spherical micelles. The solvent vapor increases the mobility of the polymer molecular chains by swelling the membrane, and consequently, some of the P4VP blocks fuse. A single SVA process is insufficient to drive the fusion of P4VP spherical structures and subsequently convert them into columnar structures perpendicular to the membrane surface as a result of strong repulsion between the PS and P4VP blocks (B). During the cooling process, chloroform vapor escapes from the inside along the direction perpendicular to the membrane as the membrane shrinks, which allows the spherical structures to aggregate along the vertical direction (C). During further SVA with programmed heating, the aggregation of spherical microdomains can propagate to the inner center of the film (E), resulting in membrane expansion (D), and shrink when cooled. After several cycles of operation, vertically oriented P4VP cylindrical microdomains are obtained (F). This experiment provides an effective orientation method for the orderly arrangement, with strong interactions between the blocks or thick membrane. Furthermore, combining thermal annealing with SVA is enabled to surmount the structural uncontrollability difficulties of high molecular weight BCP, which is due to the low mobility of molecular chains caused by chain entanglement [113].

As early as 1991, Gurovich [114] made a theoretical derivation of the force and phase behavior of BCP in an electric field, but application of electric fields in the research of BCP has stopped at the theoretical level for a while since then. This delay ended when A. Bo¨ker [115] observed the morphological changes of due to the application of DC electric field on the PS-b-PI prepared via reactive anion polymerization. In the absence of an applied electric field, the PS-b-PI self-assembled as ordered lamellar structures in the control group, while under the application of an electric field, lamellar structures of the samples underwent macroscopic orientation along the electric field lines. The application of electric field achieves long-range ordered nanostructures. Schmidt [116] verified the changes in the lamellar structure of PS-b-PI with increasing strength of the electric field. The phase space parallel to the electric field direction decreases, while the phase space perpendicular to the electric field direction increases, mainly resulting from the stretching of polymer molecular chains along the electric field lines (the mechanism is shown above in the Figure 6c). Schoberth et al. [117] used quasi in situ scanning force microscopy (SFM) capable of solvent vapor treatment of BCP despite high electric fields to characterize the phase behavior. Figure 6c below shows the SFM image of the disordered Polystyrene-b-poly (2-hydroxyethyl methacrylate)-b-poly (methyl methacrylate) S_47_H_10_M_43_ oriented along the electric field direction after the application of an electric field [118]. Subsequently, it was experimentally demonstrated that application of an electric field decreases the *T_ODT_* of PS-b-PI [119].

BCP orientation can also be regulated by the application of shear fields [120]. Unidirectional shear force is applied on the BCP membrane by contacting with suitably selected moving solid surface to induce the internal structure transformation of BCP. The shear modification of polystyrene-b-polybutadiene-b-polystyrene (SBS) and the study of its rheological behavior were carried out by Morrison et al. [121]. The structure of SBS can become oriented along the direction of shear force. Angelescu et al. [122] demonstrated that the rate of shearing plays a decisive role for obtaining the molecular alignment. Low shearing rates allow the molecular chains to arrange in an ordered manner along the shear direction, while too-high shear rates may cause a disordered arrangement of the intramolecular structures. It was then shown that for similar temperature, shear orientation requires less time compared to annealing and generates structures with other defects such as disruption of grain boundaries and absence of dislocations almost in the shear range [123]. Pujari et al. [124] observed a lamellar structure perpendicular to the substrate when the self-assembly of BCP was obtained through the application of shear stress to PS-b-PMMA during annealing. Different structures of BCP such as S, C, and L structures were oriented through the application of shearing. Davis et al. [125] investigated the morphological changes of poly(styrene)-b-poly (n-hexyl methacrylate) (PS-b-PHMA) due to application of shearing and also probed the effect of shearing on different polymer composition (see Figure 6d for a schematic diagram of the apparatus for applying shear and its results). It was finally concluded that shear causes orientation along the shear direction, and that for membranes with varying PS content, the dislocation density is maximum for the membrane containing the highest amount of PS. The ordering of shear field presents a fresh approach toward nanostructural adjustment of BCP membrane, which can achieve unprecedented structural ordering at lower temperatures and under simpler conditions [126,127].

**Figure 6 polymers-14-04568-f006:**
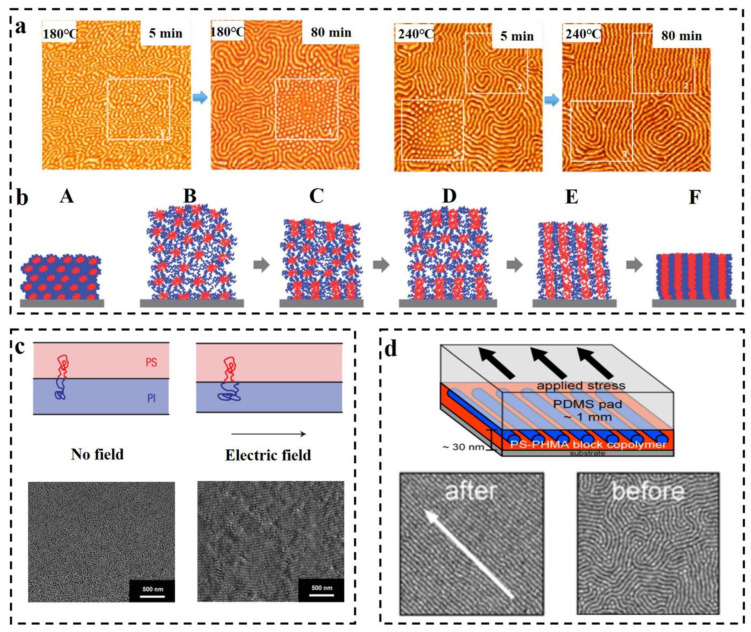
(**a**) In situ AFM images (1.25 × 1.25 μm^2^) of PS-b-PMMA obtained by thermal annealing at 180 °C and 240 °C for 5 min and 80 min, respectively [109]. (**b**) The cylindrical microdomain structure of PS-b-P4VP perpendicular to the surface is illustrated by the combination of programmed temperature rise and fall and SVA [112]. (**c**) The impact of electric field on the stretching of BCP molecular chains is illustrated and the microdomains are macroscopically oriented along the electric field direction in SEM images [116,118]. (**d**) Schematic diagram of the unidirectional shear force applied to PS-b-PHMA and TM-AFM phase images of the block nanoscale before and after shear force application. Scale bar: 500 nm [125].

#### 3.2.4. Additives for Regulating Self-Assembly Structure

Researchers also explored various types of substances that can act as electrostatic additives [128] to regulate the nanomorphology of BCP [129], especially, to obtain some morphology in narrow interval [130]. Ionic liquid (IL) is a liquid composed of only anions and cations, with high ionic conductivity, chemical stability, and high thermal stability. Ionic liquids, when doped into BCP, can selectively complex with the charged blocks and induce lyotropic phase transitions via strong electrostatic cohesion [131]. Kim et al. [132] impregnated BCP into different ionic liquids to observe the variation in morphologies. It was demonstrated that the IL, 1-ethyl-3-methylimidazolium p-toluenesulfonate, could induce the structural transformation of the BCP from L to C morphology at a particular concentration, and different ILs result in the induction of different morphologies. Later, Bennett [133] added the IL 1-ethyl-3-methylimidazolium bis(trifluoromethanesulfonyl)imide (EMIM) to PS-b-PMMA to obtain a phase transition in the following sequence: D→L→G→C→D→S. IL-induced lyotropic phase transitions of BCP correspond to the single ion-induced phase transitions, while the role of multicharged ions on the phase transitions have rarely been reported, although it is easier to obtain strong electrostatic cohesion by multicharged ions. Polyoxometalate (POM) is a class of polymetallic oxygen cluster compounds formed by pretransition metal ions linked by oxygen, which has recently been applied for controlling the BCP nanostructures [134,135]. Zhang et al. [136] observed a change of an originally ordered columnar phase structure of the block copolymer into a disordered bicontinuous phase structure via incorporating Keggin-type POM H_4_SiW_12_O_40_ (SiW) into PS-b-P4VP, which was due to strong electrostatic cohesion between the POM and P4VP chains.

## 4. Mass Transport Applications in Energy Conversion of Block Copolymers

BCPs are one of the most important materials for the preparation of nanostructures by the bottom-up self-assembly method. Self-assembly of BCPs results in well-defined nanostructures with suitable volume fractions as well as Flory–Huggins interaction parameters. These structures can be used as nanochannels’ template for mass transport. Regulation of the functional groups and structures of the channels is possible via molecular structure designing. The channel size can be tuned between 10 to 100 nm and channel density can reach up to 10^11^ cm^−2^. High-resolution nanopatterns can be widely used in the energy field, such as sensors, memory, battery separator, etc. We concentrate our focus on mass transport applications of BCP in the energy conversion.

### 4.1. Ion Selective Membranes for Concentration Cell

Use of ion-selective membranes for converting osmotic energy into electrical energy is known as reverse electrodialysis (RED) [137,138,139,140]. Concentration cells use this energy conversion system based on salinity gradient [141,142,143]. Concentration cells are expected to solve the current energy crisis [144,145] by converting the salinity-gradient energy between seawater and river water into electricity [146]. However, the performance of conventional commercial ion-selective membranes has not been satisfactory due to their high resistance and low electrical power density. The key in developing salinity-gradient-dependent energy conversion is the preparation of high-performance ion-selective membranes [147,148]. In living organisms, ion channels are mainly composed of asymmetric membrane proteins embedded in lipid layers; these channels are capable of controlling the transport of specific ions or molecules through the membrane and also participate in various complex life activities of the human body.

Researchers have been working to imitate nature in order to find high-performance membrane materials [149]. Different shapes of nanopores comparable to lipid layers on membranes have been etched by methods known as ion-path etching [150]. Then, asymmetric nanochannels can be obtained by modifying the charge on the surface of the nanochannels [151]. Surface charge inside the inner wall of channels is the major controlling factor of ion transport [152,153]. The surface charge shows ion selectivity via adsorption of counter ions and repulsion of the co-ions [154], which can be used for salinity-gradient-driven energy conversion. Due to distinctive and controllable self-assembly properties, ion-selective membranes based on BCP have been used as separators in concentration cells [155]. Researchers carried out controlled synthesis of BCP membranes with different self-assembled structure for ion transportation. The key to improving the energy conversion efficiency of membranes mainly lies in the construction of ordered [156], high-density nanochannels having asymmetric chemical composition, pore size, and surface charges [157].

The common approach toward construction of asymmetric nanochannels is to combine BCP with some pre-existing and well-studied nanochannels. For example, based on their excellent physical and mechanical properties and dimensional stability, polyethylene terephthalate (PET) membranes are often fabricated into nanopores of various shapes by ion-path etching [158]. Zhang et al. [159] added self-assembled positively charged PS_48400_-b-P4VP_21300_ onto the surface of negatively charged PET membrane. The structure of the hybridized membrane is shown in Figure 7a (left). PS_48400_-b-P4VP_21300_ self-assemble into highly ordered, hexagonally packed pores, and when the *pH* is below the *pKa*, the P4VP chains exhibit a swollen, positively charged, and hydrophilic state. Ion-path-etched PET membrane contains conical channels. In the hybrid membrane, nanochannels with asymmetric chemical structure, channel shape and size, and surface charge polarity were formed, and the resulting membrane showed anion selectivity and ultrahigh ion rectification (rectification ratio *f_rec_* up to 1075). The asymmetric structure of the hybrid membrane increases the energy conversion efficiency by eliminating the concentration polarization phenomenon to obtain a power density of 0.35 W m^−2^ in a 50-fold salinity gradient (Figure 7a (right)).

Wang’s group [160] introduced the strategy of layer-by-layer (LBL) self-assembly to regulate the channel size and improve the ion transport capacity of the channels. The PET membrane with bullet-shaped pores was immersed in a mixture of PS-b-P4VP and homopolymerized polystyrene (h-PS). Then, the BCP and homopolymer was bonded onto the surface of the PET membrane and the inner surface of the pores by employing the solvent annealing-induced nanowetting in the template (SAINT) method. The self-assembly of LBL on the surface of PET membrane and the inner surface of the channel improves the energy conversion efficiency by reducing the effective pore size. Furthermore, as shown in Figure 7b, pH value affected the stretching state of the P4VP chains introduced within the bullet PET channel, leading to pH-responsive behavior of the wettability, ion rectification, ionic conductivity and ion selectivity of this hybrid membrane. Previous studies have shown that the performance of the membranes in concentration cell systems is usually pH-responsive, while few systems have maintained good ion rectification even in multiple pH environments. Yang’s group [161] incorporated poly (tert-butyl methacrylate) (PtBuMA) into the PS-b-P2VP/PET hybrid nanochannels (the structure of the hybrid membrane is shown in Figure 7c (left)), which hydrolyzes in alkaline solution to produce negative charges, while the pyridine group in the P2VP block provide positive charges in acidic conditions. The membrane maintained a high *f_rec_* (up to 200) with changing pH values (its rectification properties are shown in Figure 7c (right)). Similarly, Wu’s group [162] employed the LBL method to graft thermally responsive PBOB-b-PNIPAM (with negative charge) into PET conical pores (schematic diagram of the synthesis process is shown in Figure 7d). The *f_rec_* increased from 1.64 to 10.66 by introduction of the polymer into the PET hole, while increasing the number of layers resulted in decreasing conductivity and *f_rec_*. As wettability and molecular conformation of PBOB-b-PNIPAM in channels depend on temperature, selective change of ion rectification and conductivity of the channels can be obtained by changing the temperature.

**Figure 7 polymers-14-04568-f007:**
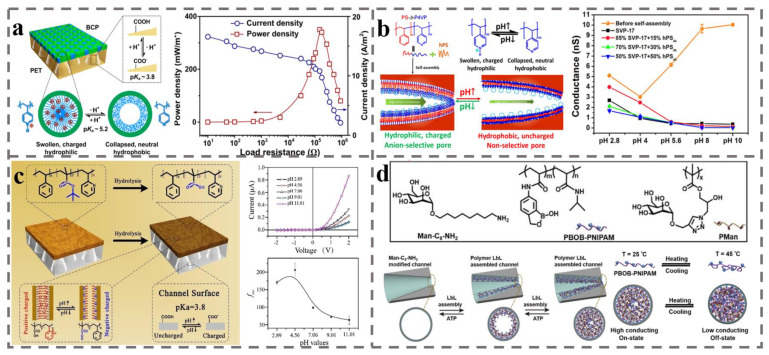
(**a**) Schematic diagram of a hybrid membrane consisting of a pH-responsive BCP membrane and a PET membrane with tapered pores obtained by ion-path etching and its permeation energy conversion efficiency [159]. (**b**) The pH-sensitive ionic conductivity of PS-b-P4VP with h-PS formed by self-assembly of molecular chain morphology with h-PS within bullet-shaped PET nanopores [160]. (**c**) The pH-responsive carboxyl groups, pyridine groups on BCP, and carboxyl groups in the PET pores enable pH-responsive, unidirectional rectification of ions in BCP-PET-laminated membrane [161]. (**d**) Molecular structures of feedstock polymers and synthetic routes to temperature-sensitive hybrid films [162].

In addition, one-dimensional (1D) and two-dimensional (2D) hybrid membranes can also be obtained when BCP is composited with 2D layered nanochannels. 2D nanomaterials have attracted the interest of researchers due to their simplicity of preparation, ease of controlling their structure and components, and possibility to extend in a large scale [163]. The unique layered stacking structure of 2D nanomaterials possesses large specific surface area and can be easily modified with functional groups. For example, MXene, with intrinsic functional groups between its layers, has received wide interest in the application of concentration cell [164]. Lin et al. [165], in an innovative method, hybridized negatively charged MXene membrane with positively charged PS-b-P2VP membrane into inhomogeneous membrane containing asymmetric structure, components, and charges. The ion selectivity experiment indicated dominant anionic selectivity for the BCP membrane with respect to MXene. This hybridization resulted in a power density of 6.74 W m^−2^ at pH = 11.

Beside these commercial BCPs, a newly designed functional BCP has also been developed. Our group [166] synthesized a UV-sensitive PEO-hv-PChal by ATRP. Introduction of o-nitrobenzyl ester as a degradable group between the two blocks resulted in breakage of this group, leading to the in situ formation of carboxylic acid groups under ultraviolet light. Self-supporting ordered nanochannels can be prepared by one-step degradation and cross-linking under the same UV light by employing PEO with good solubility as the degradation removal phase and the liquid crystal molecule containing chalone group as the continuous phase (Figure 8a). The size, density, and degree of order of the channels can be controlled by molecular weight and dispersion of the BCP. Then, we constructed asymmetric nanochannels by compounding the ordered nanochannels with AAO membrane. AAO is used by researchers in various fields of research because of its precise, stable, and un-deformed honeycomb structure with uniform pore size distribution and adjustable aperture. This asymmetric organic–inorganic hybrid membrane owned different channels and groups. Solution pH controls the ionization states of both the carboxyl group in the PChal and the hydroxyl group in the AAO pore (the pH-influenced groups within the hybridized pore channel are shown in Figure 8b). High energy conversion efficiency can be achieved over a wide pH range due to the synergistic effect of the hybrid nanochannels [167].

Although the above hybridization methods of combining BCP with other organic or inorganic membranes are able to achieve high rectification ratios or high power densities, further improvement of their performance is limited by the high internal resistance of the membrane (membrane thicknesses are often only in the micron range) and weak adhesion between the heterogeneous membranes. This motivated researchers to search for a strategy that can achieve high conversion efficiency with low internal resistance of the membrane. Zhang et al. [168] achieved innovative hybridization of two BCP membranes composed of different structural units (PEO-b-PChal and PS-b-P4VP) into an ultrathin ion-selective Janus membrane having only 500 nm thickness (the structure is shown in Figure 8c). This Janus membrane achieved excellent ion rectification due to asymmetric chemical structure, geometric aperture structure, and opposite surface charges. The results show that the P4VP chain in the channel of PS-b-P4VP membrane plays a dominant role in ion selectivity, making the hybrid membrane anion-selective. However, the operating conditions of the Janus membrane depend on the state of P4VP chains, as the performance of Janus membrane is mainly affected by the P4VP chains. Finally, this hybrid membrane was able to achieve a power density of 2.04 W m^−2^ in a seawater–river water environment at pH = 4.3.

However, the ion transport of BCP-based nanochannels is hindered due the resulting disorder and embedding of functional group distribution originating from disorder due to chaotic nature of the random coil conformation of P4VP flexible chain in the range of 1–10 nm, which makes them easy to unwind. This is a limiting factor for the further improvement of BCP-based nanochannels’ performance. Our group [169] further proposed hybridization of rigid rod-like-structured PS-b-PPLG onto the prepared PEO-b-PChal porous membrane. Poly (γ-benzyl-L-glutamate) (PBLG) is a poly(polypeptide) with stable α-helical rigid structure, and its branched chain makes it easier to modify with ionic liquids as functional groups. After the introduction of ionic liquid imidazole bromide on the α-helical rigid-structured PBLG to provide positive charges, this hybrid membrane possesses nanochannels with asymmetric surface charge and nanostructure (the structure of the asymmetric nanochannel membrane is shown in Figure 8d). The rigid PPLG segments exist in a straight chain conformation in the channel, and the arrangement is ordered in the size range of 1–10 nm, forming hierarchical ordered structures with different sizes. This method provides a strategy to improve the orderliness of the channel and reduce the internal resistance of nanochannels.

To further improve the power density of energy conversion, Li et al. [170] demonstrated a robust mushroom-shaped (with stem and cap) nanochannel array membrane with an ultrathin selective layer and ultrahigh pore density. The stem parts, negative-charged 1D channels, are prepared from the previous PEO-b-PChal self-assembly with a density of ~10^11^ cm^−2^, while the cap parts, positive-charged 3D channels, are formed by chemically grafted hyperbranched polyethyleneimine. As shown in Figure 8e, the hyperbranched polyethyleneimine cap parts as the selective layer are equivalent to tens of 1D nanochannels per stem. These robust mushroom-shaped nanochannels achieve *f_rec_* of 17.3 and power density of 15.4 W m^−2^ at a KCl solution with 500-fold salinity gradient.

**Figure 8 polymers-14-04568-f008:**
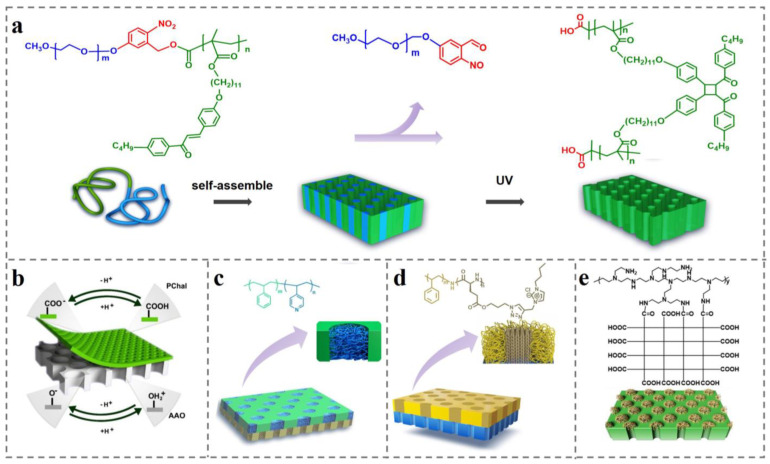
(**a**) Structural formula of PEO-hv-PChal molecule which can be degraded and cross-linked simultaneously under UV light and its self-assembly process [166]. (**b**) Schematic diagram of BCP-AAO asymmetric nanochannel membrane structure [166]. (**c**) Schematic diagram of PS-b-P4VP nanopore channel and schematic diagram of asymmetrical ultrathin Janus film hybridized with PEO-b-PChal porous membrane [168]. (**d**) Schematic diagram of the nanopore channels of PS-b-PPLG and its hybridization with PEO-b-PChal porous membrane to form a membrane with asymmetric nanopore channels [169]. (**e**) Schematic diagram of the molecular chain structure of PEO-b-PChal and polyethyleneimine and their hybridization to form mushroom-like asymmetric nanopore channels [170].

The SNIPS method also gives a new perspective for the preparation of monolayer asymmetric membrane. Koo et al. [171] combined SNIPS with a thermodynamic method for controlling the growth process of nanochannels in the membrane to achieve an asymmetric porous PS-b-P2VP membrane. The porous BCP membrane formed by the SNIPS method can spontaneously form asymmetric nanopore channels with dense nanopores on the surface and spongy and looser micropores inside. Parameters such as membrane thickness can also be controlled by thermodynamic methods. In addition, the P2VP channels can be made positively charged by quaternization, which imparts anion selectivity into the membrane and reduces the pore dimensions. Connecting a large number of salinity-gradient cells with PS-P2VP membranes can be used for the operation of small electronic devices.

Xie et al. [172] exploited a new method for the utilization of BCP in the construction of nanochannels: the nanochannels can be regulated by utilizing and modifying the different chemical properties of each block of BCP as a template. Different hydrophilic properties of PEO and PPO in triblock copolymer Pluronic F127 were used as soft templates to synthesize 1D anisotropic carbon-ordered mesoporous nanowires (CMWs). Then, dense CMWs membranes (with negative charge) were synthesized on porous AAO membranes (with positive charge) by the vacuum filtration method. In the CMWs membrane, ions can pass through the 3D interconnected channels formed by the gaps of these CMWs. Therefore, the prepared asymmetric hybrid membranes possess asymmetric chemical components, nanostructures, and surface charges, and the membrane exhibited excellent cation selectivity.

### 4.2. Ion Exchange Membranes for Fuel Cell

A fuel cell is a device capable of converting chemical energy from fuels (such as methanol, ethanol, pure hydrogen, natural gas, and gasoline) and directly oxidizing it into electrical energy through an electrochemical reaction [173,174]. It has the advantages of high efficiency, no toxic gas emission, and no pollution, and is regarded as the most promising way for generation of electricity [175,176]. For example, in hydrogen fuel cell, the basic principle is the reverse reaction of water electrolysis, where hydrogen and oxygen are supplied to the anode and cathode, respectively. After outward diffusion of hydrogen through the anode, it reacts with the electrolyte to release electrons which then reach the cathode through external load. Electrolyte diaphragm is an important part of fuel cells [177]. Its main function is to conduct ions while keeping the oxidizer and reducing agent separate [178]. The ion transport occurs through the groups present in the membrane-forming material, through the combination and separation of ions, and via forming a strip of ion channels [179]. For example, proton-exchange membranes usually have some strong electrolyte groups such as sulfonic acid radical that can easily bond with the protons [180]. Protons can easily bond with the groups and release, allowing the protons to pass through the membrane to form a current without direct contact between the positive and negative oxidizers and the fuel. Similarly, the function of cation-exchange membrane [181] and anion-exchange membrane is to allow the cations or anions to pass through, forming a current, while blocking positive and negative oxidizer and fuel. The principle is based on the selective permeability of ion-exchange membranes [182].

A proton-exchange membrane fuel cell is a new type of fuel cell with high efficiency and low pollution [183,184]. Proton-exchange membrane (PEM) [185], also known as polymer electrolyte membrane, is a very thin, rigid, plastic-like polymer material with proton conductivity that can be used to replace conventional liquid electrolytes [186,187]. The biggest challenge of PEM is to improve ionic conductivity and thermal stability without compromising the mechanical strength [188,189,190]. Inspired by the common BCP structures having rigid blocks as the skeletal support and flexible blocks as ion conductors, BCPs are usually used as precursors for nanostructured polymer materials [191].

The bicontinuous phase of self-assembled BCP has attracted much attention. There are two separate and mixed structures, the domain is throughout the material structure, it has a very high interfacial area and connectivity, and it is advantageous to the ion transport in the membrane. However, bicontinuous phase in the phase diagram has a relatively narrow interval, and it is difficult to achieve by controlling the molecular weight and proportion. Doping of other molecules to regulate the interaction forces is a good strategy to regulate the self-assembled structure of BCP. Electrostatic interaction between IL [192,193] and BCP is a promising pathway to induce BCP to form a bicontinuous phase. Morgan et al. [194] used 1-butyl-3-methylimidazolium bis (trifluoromethylsulfonyl) imide (BMITFSI) to induce the self-assembly of AB-type diblock copolymer PS-b-PEO. BMITFSI is immiscible with PS, and it enters the PEO block to induce the separation of PS-b-PEO assembly into a bicontinuous phase (the bicontinuous phase structure is shown in Figure 9a). Additionally, as shown in Figure 9a, IL also reduces the intermolecular forces between the polymer chains to enhance the fluidity of polymer chain, improves the ionic transport property of the membrane, and affects the free volume and flexibility of the membrane, improving the conductivity and electrochemical stability of the membrane.

It has also been found that in spite of IL, POM is also used to induce BCP to establish a bicontinuous phase structure. Zhang et al. [195] induced the transition of an AB-type diblock copolymer PS-b-P2VP from layered structure to bicontinuous phase structure using H_4_SiW_12_O_40_ (SiW) (the structural transformation process is shown in the top diagram of Figure 9b). POM in BCP plays the role of electrostatic morphology control system, improves the plasma conductivity, and also plays the role of nanointensifier for improving the modulus. Ultimately, as shown in the lower graph of Figure 9b, POM increased the proton conductivity *σ* and Young’s modulus of the bicontinuous structured material at room temperature by 0.1 mS cm^−1^ and by 7.4 G Pa, respectively. Zhai et al. [196] introduced H_3_PW_12_O_40_ (PW) into the synthesized ABA-type triblock copolymer PVP-b-PS-b-PVP. As shown in Figure 9c, weak electrostatic interactions between PW and PVP units induced the BCP to form a three-dimensional continuous-charged domain (for ionic conduction) and an inverse cylindrical phase structure with embedded neutral domains (for mechanical support). The nanocomposite corresponding to this configuration was shown to be a novel PEM with proton conductivity *σ* of 1.32 mS cm^−1^.

**Figure 9 polymers-14-04568-f009:**
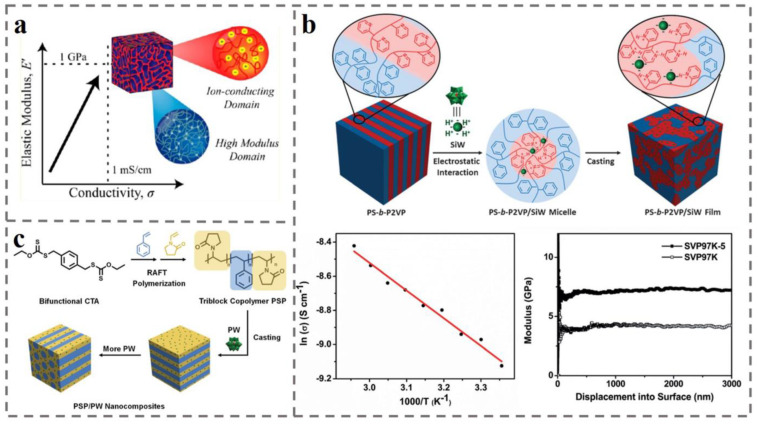
(**a**) BMITFSI induces PS-b-PEO to form a bicontinuous phase microdomain structure and enhances the elastic modulus *E*’ and conductivity *σ* of PS-b-PEO [194]. (**b**) The electrostatic interaction between SiW and PS-b-P2VP induces the formation of a bicontinuous phase favorable to proton conduction, leading to an increase in proton conductivity and Young’s modulus [195]. (**c**) PW induces PVP-b-PS-b-PVP by electrostatic interaction to form a proton-conducting continuous phase with cylindrical, mechanically enhanced phase with great proton-conduction advantage [196].

The alkaline fuel cell composed of anion exchange membrane (AEM) is similar to the PEM fuel cell in terms of mechanism and complete reaction equilibrium [197], but the electrode reaction is different from that of the PEM fuel cell, as the reaction takes place under alkaline conditions. Alkaline fuel cells have distinct advantages: the cost of fuel cell production can be reduced by some inexpensive catalysts such as iron and nickel [198,199]; liquid fuels such as methanol and ethanol that can be stored and transported easily are used [200]; and the corrosion of metal catalysts is less than that of acidic environment, prolonging the fuel cell life [201]. AEMs are the critical components of alkaline fuel cells. Their ionic conductivity and stability under alkaline environments are critical to the performance of alkaline fuel cells [202,203]. Microstructure of the polymer determines the material performance, and the design of a customized AEM with a rational structure is essential for the production of high-performance alkaline fuel cells [204,205].

Sulfonated poly (styrene-ethylene-butylene-styrene) copolymer (SEBS) is a promising diaphragm material, having high thermal and chemical stability, adjustable mechanical properties, and good proton conductivity and cost-effectiveness [206,207,208]. S-SEBS-g-MA AEM was prepared by grafting sulfonic acid and maleic anhydride to form ion channel for improvement of ion conductivity [209] (Figure 10a). From electrochemical analysis, as shown in Figure 10a (right), the modified membranes showed increased ionic conductivity, ionic exchange capacity (*IEC*), and water absorption, all of which were higher than those of the conventional commercial Nafion 117 membrane. These results suggest that modified AEMs based on BCPs such as SEBS can have comparable potential as AEMs with respect to the commercial Nafion 117.

Polyolefin-based AEMs exhibit good alkali resistance and have great potential for mass production due to their easy processability and low cost [210,211]. In the anionic polymerization, the chemical composition, molecular weight, and conformational distribution of the products can be controlled during the process of polymer synthesis. Polyisoprene-b-poly(4-methylstyrene) (SCP) AEMs with a star topology and improved properties were prepared (see Figure 10b left diagram for SCP-AEM structural formula) through anionic polymerization by Pan et al [212]. This star-shaped structure is a head-to-head cross-linked structure. The conductivity and low water absorption and alkaline stability of AEM can be significantly improved through functionalization such as bromination and quaternary ammonium near the “star-shaped”, nucleus resulting in the construction of continuous ion transport channels (AFM phase diagram is shown in Figure 10b (middle)). The *IEC* and hydroxide conductivity of the functionalized AEM can reach 2.15 mmol g^−1^ and 68.1 mS cm^−1^, respectively. With increasing current density, maximum power density of the fuel cell reaches 120.2 mW cm^−2^. This method provides a new approach to enhance the performance of AEM by taking advantage of the customizable structural properties of BCP. The conductivity and basic stability of AEM can be improved by placing the functional groups that can build continuous ion channels and the chain segments that are easily attacked by hydroxide in the inside structure of BCP by structural design.

**Figure 10 polymers-14-04568-f010:**
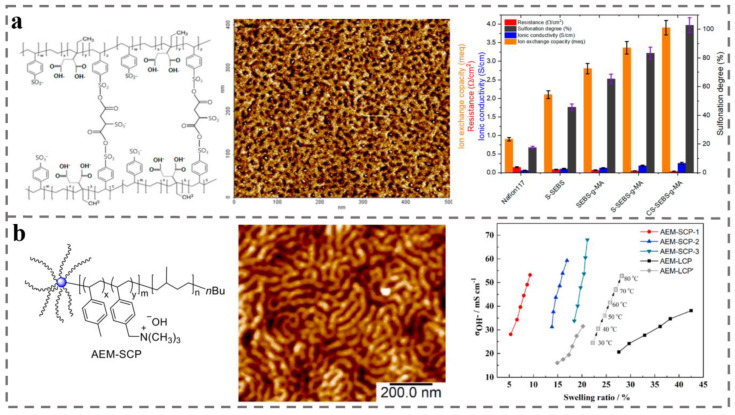
(**a**) Molecular structure formula of S-SEBS-g-MA AEM and its AFM phase image. Compared with AEM formed by grafting different molecular chains and Nafion 117, S-SEBS-g-MA AEM has the superior ionic conductivity and *IEC* performance [209]. (**b**) Molecular structure and AFM image of AEM-SCP membrane and its anion conductivity versus swelling ratio [212].

### 4.3. Battery Separator for Rechargeable Batteries (e.g., Lithium-Ion Batteries)

Among commercially implemented rechargeable batteries, high specific energy and stable cycling performance have resulted in the broad applications of lithium-ion batteries (LIBs) [213,214] in cell phones, notebook computers, electric vehicles, and grid energy storage, with prospects for further development. However, the performance and safety of lithium-ion batteries still need improvement [215,216]. As a key component, lithium ions transfer between the positive and negative electrodes during the charging and discharging process and are controlled by a battery separator, which in turn controls the efficiency of the lithium-ion battery [217,218]. An important parameter characterizing the migration of lithium ions is *t+*, which is defined as the ratio of Li^+^ migration rate to that of the migration of all ions in the electrolyte for a lithium-ion battery [219,220]. In liquid electrolytes, *t+* is normally below 1, with the general value being between 0.3 and 0.4 [221]. Traditional LIBs commonly used organic solvent-resistant, high-strength porous polyolefin membrane as a battery separator, which has good chemical stability and mechanical stability, but the thermal stability is poor. During heat build-up and rise of temperature inside the battery, the diaphragm can melt, which results in a short-circuit between the positive and negative terminals, causing an accident [222]. In recent years, there have been frequent safety accidents concerning LIBs, which seriously threaten the safety of human life and limit the application prospects of LIBs. Researchers are committed to finding ways to improve the safety of LIBs [223,224], especially by enhancing the thermal stability of the battery diaphragm to prevent accidents.

Researchers have employed polymer membranes with higher melting points and good thermal stability, such as polyimide, cellulose, and PSF, to replace traditional polyolefin materials, which has shown primary results regarding enhancement of thermal stability of LIBs. Yang et al. [225] studied PSF-b-PEG, a BCP composed of PSF and hydrophilic polyethylene glycol PEG, as a diaphragm for LIBs (diaphragm structure and lithium-ion transport process are shown in Figure 11a). This BCP has the following advantages: high thermal stability of PSF, strong affinity for the liquid organic electrolytes, and good complexation ability of PEG toward lithium salts, allowing for transport of lithium ions, which greatly improves the lithium-ion conductivity. In addition, presence of benzene ring and ether bond on the PSF chain of PSF-b-PEG provides rigidity and flexibility, performing good mechanical strength. A strong interaction force between the two blocks results in good mechanical stability for PSF-b-PEG during the formation of membrane and channels, resulting in no ion channel blockage. As is shown at Figure 11b,c, PSF-b-PEG has been tested to be comparable to commercial polypropylene Celgard 2400 membranes, both with respect to thermal stability and porosity at 380 °C, but the PSF-b-PEG membranes have additional advantages of much higher wettability, mechanical strength, and electrolyte absorption than Celgard 2400. During the tests in the temperature range of 75 °C to 150 °C, PSF-b-PEG membranes consistently maintained relatively low thermal shrinkage, ensuring the safety of LIBs. Additionally, at temperatures higher than 125 °C, the channels of PSF-b-PEG membrane will automatically close due to thermal annealing, switching “off” the working state of the battery diaphragm, and during the process of channels closure, there will not be any change of membrane size, ensuring the safety of LIBs.

Even after using thermally stable diaphragms, safety accidents cannot be eliminated at the root, and liquid organic electrolytes still have safety hazards [226]. Solid polymer lithium-ion batteries, which use solid polymer electrolytes (SPE) instead of traditional liquid electrolytes [227], are safer than traditional liquid LIBs and will gradually become the mainstream of LIBs by replacing the traditional liquid LIBs [228,229]. The ion channels of SPE should be perpendicular to the electrode surface to facilitate ion transport [230]. BCP electrolytes (BCPEs) are one of the most attractive alternatives to traditional liquid electrolytes in LIBs, as they can improve thermal and mechanical stability while maintaining ionic conductivity.

In general, rigid blocks are used as the skeleton support to provide mechanical strength, whereas flexible blocks are used to enhance ionic conductivity. PEG has been widely used in LIBs but has associated problems such as low ionic conductivity, poor mechanical strength, and narrow electrochemical window, which can be improved by reintroducing rigid blocks. Lin et al. [231] utilized cross-linking copolymerization reactions of flexible PEG blocks and rigid hexamethylene di-isocyanate trimer (HDIt) blocks in different ratios to synthesize a series of new BCPEs, named PH-BCPE, with 3D networks (see Figure 12a (left)). It has been experimentally demonstrated that the ratio of two blocks *R* (*n**_PEG_*/*n**_HDIt_*) can be a controlling factor for the performance of PH-BCPE. By increasing the proportion of flexible PEG blocks (elevation of *R* value), the ionic conductivity of PH-BCPE can be increased (as shown in Figure 12a (middle)) and the interfacial resistance with the electrode can be decreased, whereas increasing the proportion of rigid HDIt blocks (decrease in *R* value) can improve the electrochemical window (as shown in Figure 12a (right)) and mechanical strength of PH-BCPE. Finally, the ionic conductivity of obtained PH-BCPE resulted in an ionic conductivity up to 5.7·10^−4^ S cm^−1^, *t+* value of 0.49, and wide electrochemical window up to 4.65 V (vs. Li^+^/Li).

To achieve a BCPE with high mechanical stability and ionic conductivity, He et al. [232] employed RAFT polymerization to synthesize a difunctional P(DBEA-co-MA)-b-PEG which consists of the UV-cross-linkable block P(DBEA-co-MA) and suspended PEG chains (the network structure is shown in Figure 12b). The mechanical strength can be improved due to the presence of tethered double bonds in P(DBEA-co-MA) which can form a cross-linked network, while the suspended PEG molecular chains with low-crystallinity can increase molecular mobility and reduce chain entanglement to improve ionic conductivity. The interfacial resistance between the electrode and P(DBEA-co-MA)-b-PEG-SPE can be reduced from 5500 Ω cm^2^ to 100 Ω cm^2^ compared to the conventional PEO-SPE. The cycle stability of the battery is doubled, and the battery can be cycled for more than 700 h at 22 °C, showing significant improvement of durability. Using bis (trifluoromethane) sulfonimide lithium salt (LITFSI) as the ion carrier, the final value of *t+* was calculated to be 0.35.

Among the various forms of BCP, cylindrical (C) pore channels have been applied, though the lamellar (L) structures possess higher *f* values and theoretically higher ionic conductivity than C, but L has been explored very little. Liu et al. [233] employed an intermediate layer of azobenzene to synthesize a series of tablet-b-bottlebrush (TB) BCP electrolytes (Figure 12c). The liquid crystal polymer segments can form nanopores in a directional and rapid manner after solvent thermal annealing and the presence of polyethylene oxide (PEO) side chains leads to increasing electrical conductivity. The performance of electrolyte membrane with a thickness of 200 mm or more prepared by this method is superior to other spin-coated nanoscale membrane. The final synthesized SPE also possesses an ionic conductivity of 2.19 mS cm^−1^ at 200 °C, along with superior thermal stability.

**Figure 12 polymers-14-04568-f012:**
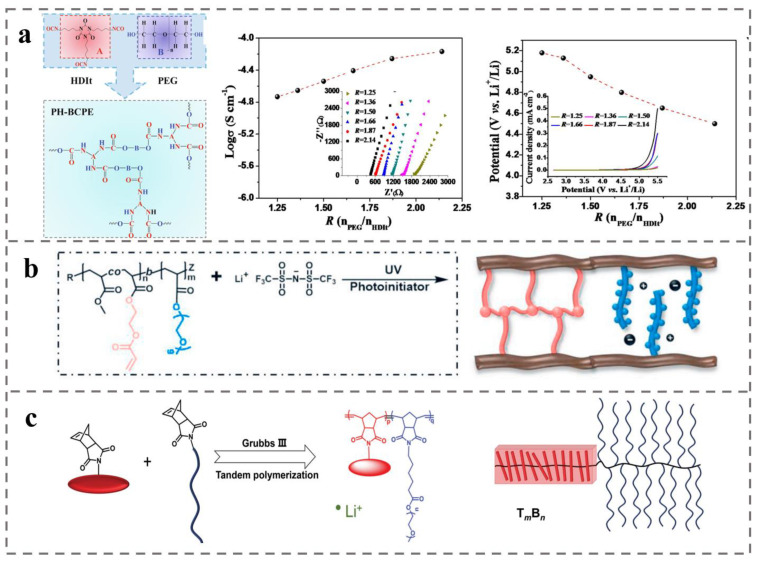
(**a**) PEG and HDIt molecules were copolymerized to obtain PH-BCPE with 3D network structure. *R* has a decisive effect on the ionic conductivity and electrochemical window of PH-BCPE [231]. (**b**) Synthesis of P(DBEA-co-MA)-b-PEG with 3D cross-linked network structure [232]. (**c**) TB-BCPEs were synthesized by tandem polymerization and have a lamellar structure favorable for ion transport [233].

## 5. Conclusions

The self-assembly behaviors of BCP can be regulated by mastering the molecular structure, solvent type, concentration of polymer solution, nonsolvent, external environmental conditions, and additives. Additionally, the self-assembly behaviors of BCP can precisely control the structure of BCP at the nanoscale. Moreover, due to their unparalleled surface-controllable and nanostructure-controllable properties, BCP can be introduced into a broad spectrum of applications. This review covers advancements in block copolymers for the optimization of energy conversion in novel batteries. The integration of BCP into new energy sources such as concentration cells, fuel cells, and lithium-ion batteries can improve energy conversion efficiency while sustaining high performance, which is unachievable with classical materials.

In spite of the attractive options provided by BCP in enhancing energy conversion efficiency, for new energy devices, there is still considerable work to be done to make them competitive with existing technologies in terms of performance and cost. The first challenge is to utilize novel materials for the synthesis of block copolymers with new and superior structures and properties. The materials currently used to synthesize block copolymers are conventional polymeric materials, and it is critical to explore new materials as well as new structures for BCPs. This becomes potentially changing as more applications are discovered. The second challenge is how to precisely tune the interface during self-assembly to tailor BCPs with absolutely precise nanostructures on demand. In many cases, different disciplines are not well-integrated, and perhaps the collision of different sciences will bring chances for precise tuning of BCPs. Finally, due to the constraints of the established technology platform and cost issues, the research on BCP-based composites is still at the primary stage of laboratory research rather than large-scale commercial application. This requires not only a change in technology, but also a long process of financial investment and training of personnel. Certainly, all of these bottlenecks are common barriers to the implementation of new technologies, and the future is still bright with the endless possibilities of BCP. We are convinced that block copolymers can enable a new era of energy conversion; the question is not if, but when.

## Figures and Tables

**Figure 11 polymers-14-04568-f011:**
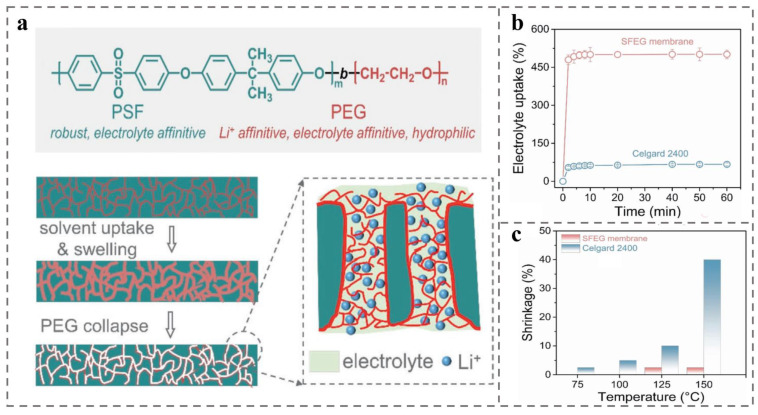
(**a**) Molecular structure of PSF-b-PEG membrane and its selective swelling application in lithium-ion transport. Compared to the Celgard 2400 diaphragm, PSF-b-PEG membrane exhibits exceptional electrolyte uptake performance (**b**) and thermal stability (**c**) [225].

## Data Availability

Not applicable.

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
