# Peer review of "Well-Defined Nanostructures by Block Copolymers and Mass Transport Applications in Energy Conversion"

_polymers, 2022, doi:10.3390/polym14214568_

Round 1

Reviewer 1 Report

This review summarized the literatures related to the construction and application of BCP-based nanostructured materials in mass transport for energy conversion application. The manuscript is well-organized. I have some minor suggestions:

1. Some conjugated BCPs are also known to self assemble into different structures and can be cited in this review: "Conjugated Block Copolymers for Functional Nanostructures." Accounts of Chemical Research55(16), pp.2224-2234.; "Bottom-up approaches for precisely nanostructuring hybrid organic/inorganic multi-component composites for organic photovoltaics." MRS Advances 5, no. 40-41 (2020): 2055-2065.

2. As most of the papers cited in the manuscript include PS block, I suggest the authors to add a table with TEM/SEM image and corresponding schematics to summarize the self assembled nanostructures from PS-based BCPs.

3. I suggest the authors to also add a perspective or future direction section at the end of the review and mention in detail what are the aspects that still need development in this field.

Author Response

Dear Reviewer

Thank you very much for your comments on our manuscript. We have answered questions in detail and have amended the manuscript carefully. All changes are highlighted in the revised manuscript in yellow underline.

This review summarized the literatures related to the construction and application of BCP-based nanostructured materials in mass transport for energy conversion application. The manuscript is well-organized. I have some minor suggestions:

Comment 1. Some conjugated BCPs are also known to self assemble into different structures and can be cited in this review: "Conjugated Block Copolymers for Functional Nanostructures." Accounts of Chemical Research, 55(16), pp.2224-2234.; "Bottom-up approaches for precisely nanostructuring hybrid organic/inorganic multi-component composites for organic photovoltaics." MRS Advances 5, no. 40-41 (2020): 2055-2065.

Response: Thank you for your suggestion, we have cited these articles as Ref. [18] and Ref. [21] respectively.

Comment 2. As most of the papers cited in the manuscript include PS block, I suggest the authors to add a table with TEM/SEM image and corresponding schematics to summarize the self assembled nanostructures from PS-based BCPs.

Response: Thanks for your suggestion. The superiority of block copolymers over other polymers is that different polymer blocks can be designed to provide block copolymers with the excellent properties of different blocks. PS is frequently used as the supporting skeleton part in block copolymers due to its high transparency, high stiffness, and high glass transition temperature. However, the structure of PS-based BCP often depends on other blocks complexed with PS, and it is difficult for us to summarize the description of PS-based BCPs. It will change the structure of our whole paper. We have carefully considered your suggestions, but since this review focuses on the structure of block copolymers, the factors influencing the structure, and the applications of block copolymers rather than on a concrete category of block copolymers. Hence, we have not added this section to the manuscript.

Comment 3. I suggest the authors to also add a perspective or future direction section at the end of the review and mention in detail what are the aspects that still need development in this field.

Response: Thanks to your suggestion, we have inserted some perspectives in the conclusion section. (Lines 891-908)

“In spite of the attractive options provided by BCP in enhancing the energy conversion efficiency, for new energy devices, there is still considerable work to be done to make them competitive with existing technologies in terms of performance and cost.The first challenge is to utilize novel materials for the synthesis of block copolymers with new and superior structures and properties. The materials currently used to synthesize block copolymers are conventional polymeric materials, and it is critical to explore new materials as well as new structures for BCPs. This is potentially changing as more applications are discovered. The second challenge is how to precisely tune the interface during self-assembly to tailor BCPs with absolutely precise nanostructures on demand. In many cases, different disciplines are not well integrated, and perhaps the collision of different sciences will bring chances for precise tuning of BCPs. Finally, due to the constraints of the established technology platform and cost issues, the research on BCP-based composites is still at the primary stage of laboratory research rather than large-scale commercial application. This requires not only a change in technology, but also a long process of financial investment and training of personnel. Certainly, all of these bottlenecks are common barriers to the implementation of new technologies, and the future is still bright with the endless possibilities of BCP. We are convinced that block copolymers can enable a new era of energy conversion; the question is not if, but when.”

Reviewer 2 Report

This manuscript focuses on nanostructured systems made up of block copolymers and analyzes the factors that affect the related self-assembly processes. Bottom-up synthesis of nanoscale polymer particles via self-organization is a fruitful approach to obtaining novel materials with tailored smart properties. Such nanosystems are demanded for a variety of practical applications and the authors pay major attention at copolymer-based mass transport membranes and their energy conversion applications.

The review includes a considerable number of up-to-date reference papers, which are published in high-ranked scientific journals.

The manuscript is thoroughly written and its review information is logically related to appropriate references.

This review can be a valuable contribution to the field provided that the authors address the following comments:

Major comments:

1. Please state the aim of the review more clearly in the last paragraph of the Introduction section.

2. The quality of the manuscript figures should be improved. It is almost impossible to distinguish small details in Fig. 1, 3-5, 7, 10, and 11. We recommend the authors to revise these figures and provide their higher resolution versions. This recommendation is also valid for all the other figures, if applicable. Another option is to enlarge smaller parts of the figures (e.g. Fig. 7c) in the revised manuscript.

3. The authors are recommended to carefully re-consider the use of the terms “microstructure, microphase, and etc. In the manuscript, nanoscale phenomena are discussed, such as self-assembly at the level of polymer blocks, templates, and the related nanostructures (lines 15 and 39 in the Introduction, for example, also lines 41,42,49, and so on). It is supposed to be more appropriate to replace them with the relevant “nano-“ terms, such as “nanostructure” or “nanophase”.

4. The authors use mixed terms “Microscopic nanostructures” (Line 138), “Microscopic nanomorphology” (Line 452), or “microdomains at the nanoscale” (Line 879). Nanostructures are non-microscopic by default and we should avoid the terms “micro-“ while discussing nanoscale phenomena or organized systems. The authors are recommended to clearly distinguish the nanoscale (1-100 nm) and microscale (usually 1-100 µm) levels in their manuscript to avoid possible confusion. If such terms as “microscopic nanostructures” or similar ones should be kept in the revised manuscript, the authors are recommended to justify their use.

5. In the Section 3.1, the authors discuss BCS self-assembly in general and demonstrate a broad variety of nanostructures. In further sections 3.2.1-3.2.4, however, the authors focus mostly on BCP self-organization that is relevant to formation of nanoscale components of membranes. To enhance the manuscript readability, it is recommended to justify the transition from general self-organized structures in Section 3.1 to more specific nanostructures that can be used for fabrication of membranes (Sections 3.2.1-3.2.4)

6. The Conclusion should be revised in a more logical and coherent way. In the provided manuscript, authors start their Conclusion with self-assembly behaviors of BCP, then mention energy conversion applications, then proceed to regulating BCP structure, and finally return to applications. The authors are recommended to modify the Conclusion in a logical sequence: structure – self-assembly – properties – applications of block copolymers.

 The concluding phrase “In spite of the attractive options provided by BCP in enhancing the energy conversion efficiency, but constraints of established technology platforms and cost issues mean successful commercial adoption of BCP in these fields is still waiting to see the first ray of light” in Lines 889-892 is confusing and hard to understand, please revise it in the modified Conclusion.

Minor comments:

1. Some paragraphs are hard to read because of their size: lines 149-178, lines 259-301, and lines 362-397. The authors are recommended to revise long paragraphs and split them into smaller text sections.

2. The authors are recommended to revise minor punctuation typos in the manuscript.

3. Lines 73 and 477: do the authors mean “templates”?

4. Line 81 – do the authors mean “in the macromolecular length SCALE”?

5. Lines 215-219. The phrase is confusing and hard to read. Please revise.

6. Line 323 – the title of the Section 3.2.2. is confusing. Do you mean “Non-solvent APPROACHES for constructing nanostructure”? Please revise.

7. Line 419 “to induce the internal structure of BCP”. Is it more appropriate to replace it with “to induce internal structural transformations of the BCP” similar to the text in line 459?

8. Line 450 “Additive for regulating self-assembly structure” is recommended to be replaced with “Additives for regulating self-assembly structure”.

9. In lines 498-500, the authors use the term “obtain” several times that makes the phrase confusing. Please revise.

10. The phrase in lines 739-741 is a bit confusing (what is a key component of fuel cells?). Please revise for a better readability.

11. Line 789 – LIB or LIBs?

12. Lines 824 and 827 – why “sPE” and not “SPE”? Please clarify or revise.

Author Response

Dear Reviewers

Thank you very much for your comments on our manuscript. We have answered questions in detail and have amended the manuscript carefully. All changes are highlighted in the revised manuscript in yellow underline.

This manuscript focuses on nanostructured systems made up of block copolymers and analyzes the factors that affect the related self-assembly processes. Bottom-up synthesis of nanoscale polymer particles via self-organization is a fruitful approach to obtaining novel materials with tailored smart properties. Such nanosystems are demanded for a variety of practical applications and the authors pay major attention at copolymer-based mass transport membranes and their energy conversion applications.

The review includes a considerable number of up-to-date reference papers, which are published in high-ranked scientific journals.

The manuscript is thoroughly written and its review information is logically related to appropriate references.

This review can be a valuable contribution to the field provided that the authors address the following comments:

Major comments:

Comment 1. Please state the aim of the review more clearly in the last paragraph of the Introduction section.

Response: Thanks for your comment, we have added the purpose of this review at the last paragragh of the Introduction section.

“We hope that this review can provide a theoretical basis for how to regulate the nanostructure of polymers and offer ideas to further promote the application of block copolymers in new energy fields.”

Comment 2. The quality of the manuscript figures should be improved. It is almost impossible to distinguish small details in Fig. 1, 3-5, 7, 10, and 11. We recommend the authors to revise these figures and provide their higher resolution versions. This recommendation is also valid for all the other figures, if applicable. Another option is to enlarge smaller parts of the figures (e.g. Fig. 7c) in the revised manuscript.

Response: Thanks for your comments. We have magnified Figure 1, 3-5, 7, 10, and 11 appropriately in the revised manuscript.

Comment 3. The authors are recommended to carefully re-consider the use of the terms “microstructure, microphase, and etc. In the manuscript, nanoscale phenomena are discussed, such as self-assembly at the level of polymer blocks, templates, and the related nanostructures (lines 15 and 39 in the Introduction, for example, also lines 41,42,49, and so on). It is supposed to be more appropriate to replace them with the relevant “nano-“ terms, such as “nanostructure” or “nanophase”.

Response: Thanks for your comments, we have replaced the "microstructure" with "nanostructure" (Line15, 39, and so on). But the meaning of "micro" in "microphase separation" and "micro domain" is relative to the macro, not the microscopic scale defined in terms of size. But the concept of "microphase separation" of block copolymers is broadly utilized in the literature on block copolymers, e.g., Refs. 1, 15 - 18 and 32, and so on.

Comment 4. The authors use mixed terms “Microscopic nanostructures” (Line 138), “Microscopic nanomorphology” (Line 452), or “microdomains at the nanoscale” (Line 879). Nanostructures are non-microscopic by default and we should avoid the terms “micro-“ while discussing nanoscale phenomena or organized systems. The authors are recommended to clearly distinguish the nanoscale (1-100 nm) and microscale (usually 1-100 µm) levels in their manuscript to avoid possible confusion. If such terms as “microscopic nanostructures” or similar ones should be kept in the revised manuscript, the authors are recommended to justify their use.

Response: Thank you for your comments, we have revised these terms. (Line135, 453, and so on)

Comment 5. In the Section 3.1, the authors discuss BCS self-assembly in general and demonstrate a broad variety of nanostructures. In further sections 3.2.1-3.2.4, however, the authors focus mostly on BCP self-organization that is relevant to formation of nanoscale components of membranes. To enhance the manuscript readability, it is recommended to justify the transition from general self-organized structures in Section 3.1 to more specific nanostructures that can be used for fabrication of membranes (Sections 3.2.1-3.2.4)

Response: Thanks to your comments, we have added some transitional content between these two parts of the manuscript. (Lines 132-134 and lines 183-186)

“The structural morphology of BCP films is our main interest. The self-assembly of BCP film is affected by many factors [43-45], which can be divided into molecular structure factors and assembly condition factors.”

“Normally, in addition to the interactions between polymer molecules, external film formation conditions can also affect the microstructure. Hence, the type of solvent, polymer solution concentration, non-solvent, additives and external field conditions during the film formation process can affect the structure of BCP.”

Comment 6. The Conclusion should be revised in a more logical and coherent way. In the provided manuscript, authors start their Conclusion with self-assembly behaviors of BCP, then mention energy conversion applications, then proceed to regulating BCP structure, and finally return to applications. The authors are recommended to modify the Conclusion in a logical sequence: structure – self-assembly – properties – applications of block copolymers.

 The concluding phrase “In spite of the attractive options provided by BCP in enhancing the energy conversion efficiency, but constraints of established technology platforms and cost issues mean successful commercial adoption of BCP in these fields is still waiting to see the first ray of light” in Lines 889-892 is confusing and hard to understand, please revise it in the modified Conclusion.

Response: Thank you for your comments, we have revised the conclusion according to the logic of " structure – self-assembly – properties – applications of block copolymers " and made some outlooks.

“The self-assembly behaviors of BCP can be regulated by mastering the molecular structure, solvent type, concentration of polymer solution, non-solvent, external environmental conditions and additives. And the self-assembly behaviors of BCP can precisely control the structure of BCP at the nanoscale. Moreover, due to their unparalleled surface-controllable and microstructure-controllable properties, BCP can be introduced into a broad spectrum of applications. This review covers advancements in block co-polymers for optimization of energy conversion in novel batteries. The integration of BCP into new energy sources such as concentration cells, fuel cells, and lithium-ion batteries can improve energy conversion efficiency while sustaining high performance which is un-achievable with classical materials.”

“In spite of the attractive options provided by BCP in enhancing the energy conversion efficiency, for new energy devices, there is still considerable work to be done to make them competitive with existing technologies in terms of performance and cost.The first challenge is to utilize novel materials for the synthesis of block copolymers with new and superior structures and properties. The materials currently used to synthesize block copolymers are conventional polymeric materials, and it is critical to explore new materials as well as new structures for BCPs. This is potentially changing as more applications are discovered. The second challenge is how to precisely tune the interface during self-assembly to tailor BCPs with absolutely precise nanostructures on demand. In many cases, different disciplines are not well integrated, and perhaps the collision of different sciences will bring chances for precise tuning of BCPs. Finally, due to the constraints of the established technology platform and cost issues, the research on BCP-based composites is still at the primary stage of laboratory research rather than large-scale commercial application. This requires not only a change in technology, but also a long process of financial investment and training of personnel. Certainly, all of these bottlenecks are common barriers to the implementation of new technologies, and the future is still bright with the endless possibilities of BCP. We are convinced that block copolymers can enable a new era of energy conversion; the question is not if, but when.”

Minor comments:

Comment 1. Some paragraphs are hard to read because of their size: lines 149-178, lines 259-301, and lines 362-397. The authors are recommended to revise long paragraphs and split them into smaller text sections.

Response: Thank you for your comment, we have split these long paragraphs into several smaller ones. (Lines 147-176, lines 261-303, and lines 363-398)

Comment 2. The authors are recommended to revise minor punctuation typos in the manuscript.

Response: Thank you for your comment, we have critically corrected the punctuation typos in the manuscript.

Comment 3. Lines 73 and 477: do the authors mean “templates”?

Response: Thank you for your comment, we have revised in the manuscript.

Comment 4. Line 81 – do the authors mean “in the macromolecular length SCALE”?

Response: Thank you for your comment, we have inserted "scale" in the manuscript. (Line 83)

Comment 5. Lines 215-219. The phrase is confusing and hard to read. Please revise.

Response: Thank you for your comment, we have revised the phrase to make it more readable. (Lines 217-221)

“When polymers are dissolved in a mixture of solvents with different solubilities and volatilities, the preferential evaporation of one solvent leads to a concentration gradient within the BCP. This concentration gradient leads to phase separation of the BCP, the so-called evaporation-induced phase separation (EIPS) [58] of the solvent.”

Comment 6. Line 323 – the title of the Section 3.2.2. is confusing. Do you mean “Non-solvent APPROACHES for constructing nanostructure”? Please revise.

Response: Thank you for your comment, we have added the “approaches” to the title of the Section 3.2.2.

Comment 7. Line 419 “to induce the internal structure of BCP”. Is it more appropriate to replace it with “to induce internal structural transformations of the BCP” similar to the text in line 459?

Response: Thank you for your comment, we have replaced “to induce the internal structure of BCP” with “to induce internal structural transformations of the BCP”. (Line 420)

Comment 8. Line 450 “Additive for regulating self-assembly structure” is recommended to be replaced with “Additives for regulating self-assembly structure”.

Response: Thank you for your comment, we have modified the correct plural form of the noun. (Line 451)

Comment 9. In lines 498-500, the authors use the term “obtain” several times that makes the phrase confusing. Please revise.

Response: Thank you for your comment, we have converted the sentence to a more understandable form. (Lines 498-500)

“Then, asymmetric nanochannels can be obtained by modifying the charge on the sur-face of the nanochannels.”

Comment 10. The phrase in lines 739-741 is a bit confusing (what is a key component of fuel cells?). Please revise for a better readability.

Response: Thank you for your comment, we have highlighted the central meaning of the phrase. (Lines 739-741)

“AEMs are the critical components of alkaline fuel cells. Their ionic conductivity and stability under alkaline environment are critical to the performance of alkaline fuel cells [202, 203].”

Comment 11. Line 789 – LIB or LIBs?

Response: Thank you for your comment, we have converted "LIBs" to "LIBs". (Line 789)

Comment 12. Lines 824 and 827 – why “sPE” and not “SPE”? Please clarify or revise.

Response: Thank you for your comment, we have converted "sPE" to "SPE". (Line 828, line 831, line 854, line 858, line 859, and line 872)

Round 2

Reviewer 2 Report

The authors addressed all my recommendations and concerns in the revised manuscript.

The review will be a valuable contribution to the Special Issue field.